# Eating Habits and Their Association with Weight Status in Chinese School-Age Children: A Cross-Sectional Study

**DOI:** 10.3390/ijerph17103571

**Published:** 2020-05-20

**Authors:** Minghui Sun, Xiangying Hu, Fang Li, Jing Deng, Jingcheng Shi, Qian Lin

**Affiliations:** 1Department of Nutrition Science and Food Hygiene, Xiangya School of Public Health, Central South University, 110 Xiangya Rd., Changsha 410078, China; sun.1234@csu.edu.cn (M.S.); hxyybyq@163.com (X.H.); 2Department of Epidemiology and Health Statistics, Xiangya School of Public Health, Central South University, 110 Xiangya Rd., Changsha 410078, China; lifang198712@hotmail.com (F.L.); jingdeng@csu.edu.cn (J.D.); shijch@csu.edu.cn (J.S.)

**Keywords:** school age, eating habits, weight status, overweight and obesity, Chinese

## Abstract

Background: Poor eating habits increase children’s risk of chronic diseases such as overweight and obesity, adult diabetes, and cardiovascular and cerebrovascular diseases. Therefore, this study aimed to examine the eating habits of school-age children and their impact on children’s body mass index. Methods: Multistage cluster sampling was used to sample 2224 students in grades 4 to 6 from 16 primary schools in Changsha. A self-designed questionnaire was used to collect general demographic, eating habit, and snack consumption data from school-age children. Height and weight were measured so that the body mass index Z-score could be calculated and evaluated according to the 2007 World Health Organization’s Body Mass Index (BMI) reference standard. Results: The prevalence rates of overweight and obesity in school-age children were 17.0% and 8.3%, respectively. Boys had higher rates of overweight and obesity than girls (19.9% vs. 13.6%, 12.9% vs. 3.0%, respectively, *p* < 0.05). Among the total population of children, 71.6% reported picky eating habits, and 55.1% had late-night snacks. Children who skipped breakfast (OR 1.507 and 95% CI 1.116~2.035) and ate puffed food (OR 1.571 and 95% CI 1.170~2.110) were more likely to be overweight/obese. Conclusions: The eating habits of school-age children are closely related to their weight status. Poor eating habits can be risk factors for overweight and obesity. The dietary management of children should be strengthened so that they develop good eating habits and the incidence of overweight and obesity in school-age children decreases.

## 1. Introduction

In recent decades, the prevalence of overweight and obesity among children has increased dramatically worldwide and has become a major challenge in public health. Research has shown that the global average body mass index (BMI) and obesity rates of children and adolescents have increased from 1975 to 2016, especially in low- and middle-income countries [1]. At present, there are 1.5 billion overweight people in the world, of which 31% are children and adolescents; 500 million people are obese, of which 6% to 8% are children [2].The prevalence of obesity and overweight among children and adolescents in China reached 7.1% and 12.2%, respectively, in 2013, and the prevalence is growing rapidly [3]. Wang and other researchers have estimated that if this situation is not improved, the prevalence of obesity in children and adolescents will likely reach 30% by 2030 [4]. Excess body mass is a direct causative factor of endothelial dysfunction, which is also associated with increased risk for numerous chronic diseases, such as adult diabetes, hypertension, ischemic heart disease, and stroke [5,6]. Childhood obesity is an important predictor of adult obesity [7,8], and therefore early intervention for children is very significant to solve this public health problem.

Studies have shown that although genetic factors play a crucial role in obesity tendencies, 82.6% of obese children have a family history of dyslipidemia and these disorders are high risk factors for childhood obesity [9]. However, further study is still needed with regard to whether parental dyslipidemia is a risk sign of offspring obesity [10]. Changes in environmental and behavioral factors also affect the risk of obesity, such as a poor diet and low levels of physical activity. Currently, diet is considered an important cause of overweight and obesity [11,12]. Poor eating habits, such as high-fat and high-energy food intake, picky eating, eating too fast, and eating while watching television have been shown to be risk factors for overweight and obesity [12].

Childhood is a critical period for individual intelligence and physical development and also for eating habit formation [13]. Eating habits developed during childhood directly affect those implemented during adulthood, and behaviors and have a vital impact on adult health [14]. In China, the eating habits of urban children are generally problematic, as the incidence of dietary problems is as high as 25%~40%, and these problems seriously affect their physical and mental health [15]. The intake frequency of sugary drinks, fried food, and snacks was high, whereas the report rate of regular breakfast and eating out was relatively lower among 13–15-year-old children [16]. Evidence suggests that overeating high-energy foods, especially saturated fats and refined carbohydrates, can lead to obesity and increase the risk of chronic noncommunicable diseases [17]. Research finds that high sugar-sweetened beverage intake frequency is associated with smoking, irregular meal intake, and higher serum uric acid in adolescents [18]. Serum uric acid is a chance factor for childhood and adolescent chronic kidney disease [19]. In addition, up to 60% of Chinese children are picky eaters [20]. Picky eating is an important cause of malnutrition (undernutrition or over nutrition), food preferences and dietary habits established by picky eaters during childhood can persist into adulthood [21]. It is of great significance to identify people’s poor eating habits early in life and implement interventions in a timely manner for the healthy growth of children and to prevent the occurrence of related diseases. Therefore, this study aims to investigate the relationship among eating habits, physical activity, and weight status in Chinese school-age children. Findings from this study could provide a basis for implementing relevant interventions to reduce the occurrence of poor eating habits in children.

## 2. Materials and Methods

### 2.1. Sample and Design

A cross-sectional study was conducted in Changsha City of Hunan Province, China from March to April 2012. Multistage cluster sampling was used to randomly select 16 primary schools from six districts and two counties. Two primary schools were randomly selected from each district, and then one class was randomly selected from grade 4, grade 5, and grade 6, in each selected school. All the students in the classes participated in the investigation. Participants and guardians signed informed consent forms before the investigation, and all information was kept strictly confidential. The PASS software package (Windows version 11.0, NCSS, LLC, Kaysville, UT, USA) was used to calculate the sample size. We hypothesized that the incidence of overweight and obesity of school-age children in Changsha city was 17% and the allowable error was 2%. Therefore, the sample size required was 1355. Considering a valid response of 70%, a total of 1935 participants should be recruited at least.

### 2.2. Data Collection

#### 2.2.1. Questionnaires

The investigators were trained uniformly. After a general explanation and clear instructions were provided with the help of the headteacher, all participants were asked to fill out self-report questionnaires comprising three sections.

The first section addressed sociodemographic information, including the participants’ sex, age, the type of area in which they live, whether they are an only child, whether their family is a single-parent family, the type of schooling they attend (boarding school or another type of schooling), snack costs (the median of this parameter before the survey was 5 RMB), and physical activity (duration per day) (note, for the Chinese currency, renminbi (RMB) 1 = USD $0.142, in 2019). The second section addressed students’ eating habits, such as skipping breakfast, choosing soft drinks to quench their thirst (soft drinks cola or sprite, mineral water, juice, tea drinks and so on), eating while watching television, picky eaters, having late-night snacks, and eating meals at irregular times. The responses for these eating habits were either “yes” or “no”. The participants were also asked to provide the frequency at which they ate out and consumed fast food (KFC, McDonald’s, etc.) in the past week, with the following response choices: never, 1–3 times/month, once a week, and ≥2 times/week. These responses were categorized as either “once a week or less” or “≥2 times/week”. For eating speed, the response options ranged from “≤10 min” to “>10 min”. The third section was used to collect the types of snacks consumed more than three times a week by school-age children during the past week.

#### 2.2.2. Anthropometric Measurements

All the participants took off their coats and shoes before their body height and weight were measured. Body height was measured using a portable stadiometer (Shanghai Bedelneng Industrial Co., Ltd., Shanghai, China), with an accuracy of 0.1 centimeters (cm). Weight was measured using a Xiangshan BR2017 weighing scale (Guangdong Xiangshan Weighing Instrument Group Co., Ltd., Guangdong, China), with an accuracy of 0.5 kilograms (kg). There was strict quality control in the measurement process, and the mean value for two times of measurement was taking as the final report result. The BMI scores were calculated from the measured data using the following formula: BMI = weight (kg)/height (m) squared. Weight status was classified according to the 2017 WHO growth references for individuals aged 5–19 years (BMI Z-score) [22] as follows: severe thinness (<−3 SD), thinness (<−2 SD), normal (−2 SD~+1 SD), overweight (>+1 SD), and obesity (>+2 SD).

### 2.3. Statistical Analysis

Double and parallel data entry was performed using EpiData 3.1 (EpiData Association, Odense, Denmark), and the data were checked logically. The SPSS 24.0 software statistical package (IBM Corp., Armonk, NY, USA) was used to analyze and process the data. Chi-square tests and descriptive statistics were used to assess the general demographic characteristics, eating habits, and snack consumption data. Logistic regression was used to analyze the factors affecting weight status. A *p*-value < 0.05 was considered statistically significant.

## 3. Results

### 3.1. General Information of the Participants

A total of 2224 school-age children were investigated, and the final valid sample population included 2185 children (Table 1 and Table 2). Of the 2185 children, 53.7% were male (n = 1174), and 46.3% were female (n = 1011). The average age was 10.83 ± 0.993 years. Over 60% of the children lived in an urban area (n = 1418). More than half (n = 1203, 57.5%) of them were an only child, the only child of boys was significantly higher than that of girls (33.8% vs. 23.7%). The percentages of children in single-parent families and boarding schools were low, as they were only 8.9% and 6.4%, respectively. For most of the (n = 1391, 66.2%) children, the snack cost was less than 5 RMB, and nearly half (n = 1081, 50.6%) of them spent more than one hour performing physical activity per day. In this study, the proportions of children with overweight and obesity were 17.0% and 8.3%, respectively (Figure 1). As shown in Table 1 and Table 2, boys were more likely to be overweight/obese than were girls (32.8% vs. 16.5%, *p* < 0.05). For both boys and girls, only children (35.7% vs. 27.0% and 20.2% vs. 12.3%, *p* < 0.05) and living in an urban area (39.0% vs. 21.9% and 19.3% vs. 11.1%, *p* < 0.05) were more likely to be overweight/obese. There were significant differences between normal-weight and overweight/obese boys for age, attending boarding school, and physical activity duration per day (*p* < 0.05).

### 3.2. Eating Habits

The eating habits of the school-age children with different weight statuses are shown in Table 3. Overweight/obese participants were generally not picky eaters; 71.6% of the school-age children were picky about food, whereas 28.4% were not (*p* < 0.001). Furthermore, compared to the children who ate breakfast daily, significantly more overweight/obese children skipped breakfast (44.4% vs. 23.8%). Most of them (1477, 69.3%) chose plain boiled water to quench their thirst, while overweight/obese children chose carbonated drinks (27.0%) or mineral water (33.7%) when thirsty (*p* = 0.039). There was no significant difference among the children in different weight statuses in terms of the frequency of eating out and fast food consumption (KFC, McDonalds, etc.), ate while watching television, eating speed, eating meals at irregular times, and late-night snacks.

Results from the frequency of eating fast food (KFC, McDonalds, etc.) in school-age children by BMI categories are presented in Figure 2. The results show that as compared with normal-weight and overweight children, the proportion of eating fast food more than two times/week was higher among obese children.

### 3.3. Types and Percentages of Snacks

As shown in Figure 3, the snacks that were most frequently consumed by school-age children were biscuits/cakes (48.4%), followed by candy/chocolate (40.1%), spicy snacks (21.4%), soda/drinks (20.6%), and puffed food (20.0%). The percentages of girls and boys eating candy/chocolate and biscuits/cakes were roughly equivalent, but that of girls was slightly higher (51.0% vs. 49.0%, 52.6% vs. 47.4%, respectively, *p* < 0.01). Boys were more likely to consume soda/drinks (65.2% vs. 34.8%, *p* < 0.001), spicy snacks (61.6% vs. 38.4%, *p* < 0.001) and puffed food (56.5% vs. 43.5%). Additionally, another 5% of children chose other kinds of snacks. Our study indicated that children who ate puffed food more than three times a week were more likely to be overweight/obese than were those who ate puffed food fewer than three times a week (31.4% vs. 23.5%, *p* < 0.01).

### 3.4. Factors Related to Children’s Weight Status

Logistic regression analysis was conducted to assess factors of weight status among school-age children (Table 4). The model was adjusted for age, whether the children were in single-parent families, schooling method, physical activity duration, and all eating habits except for skipping breakfast, picky eaters, and eating four types of snacks (candy/chocolate, biscuits/cakes, soda/drinks, and spicy snacks). We found that the odds of being overweight/obese were estimated to be 1.507 times higher for children who skipped breakfast than for children who ate breakfast daily (OR 1.507, 95% CI 1.116~2.035). The possibility of overweight/obesity was estimated to be 1.571 times higher for children who ate puffed food more than three times a week than for children who ate puffed food fewer than three times a week (OR 1.571, 95% CI 1.170~2.110). Last, to our surprise, picky eaters were less likely to be overweight/obese (OR 0.559, 95% CI 0.427~0.733) than non-picky eaters.

## 4. Discussion

In this study, we found that the total prevalence of overweight and obesity among school-age children in Changsha was 25.3%, which was considerably higher than the prevalence reported in “The National Survey on Students Physical Fitness and Health in 2014” (19.4%) [23] and higher than that in Chongqing [24] (overweight rate 11.8% and obesity rate 7.5%) and Chengdu [25] (overweight rate 10.34% and obesity rate 6.59%); however, it was lower than that in Jinan [26] (overweight rate 21.0% and obesity rate 19.9%). Since 2007, the prevalence of overweight and obesity among school-age children in Changsha has increased by 7.4% [27]. Compared with girls, boys were more likely to be overweight/obese (32.8% vs. 16.5%), which is consistent with other findings [28,29]. Boys are more active, tend to eat faster and larger amounts of food, and have more difficulty feeling full [30]. For girls, due to family-related factors, social factors, and other factors, they pay more attention to their weight status, diet, and lifestyle as they age [31]. We also found that the prevalence of overweight and obesity in urban areas was much higher than that in rural areas (19.4% vs. 12.5%, 10.3% vs. 4.6%, respectively). Due to the differences in living environment and household economic level, it is easier for urban children to obtain high-energy and high-fat food, while their level of physical activity is lower than that of rural children [32]. The only children were more likely to be overweight/obese than were children with sibling(s). Previous studies have shown that the rate of overweight or obesity among Chinese only children aged 6–18 years doubled from 2000 to 2011 (6.6% to 16.5%). In part due to the one-child policy, parents of only children tend to overprotect their children, so they perform less housework and other physical activities, resulting in an increased risk of overweight and obesity [33]. With the rapid development of the economy and society, overweight and obesity have become urgent public health issues worldwide. Diet and physical activity interventions are necessary to address these issues in children and adolescents. The American Heart Association recommends physical exercise in obesity because it helps reduce the risk factors of cardiac metabolism such as metabolic syndrome, dyslipidemia, hypertension, cardiovascular disease, insulin resistance, and inflammation [34]. At present, schools in China are focused on increasing students’ level of physical activity by implementing interventions such as requiring “outdoor sports for one hour”, “three lessons, two sessions of exercise and two activities”, and “three kinds of ball games and ice and snow sports on campus”, but they lack effective interventions to improve students’ eating habits [35].

Childhood is a crucial period for children’s growth and development. Good eating habits can improve children’s cognitive ability and behavior [36] and also reduce the risk of overweight and obesity. This is a vital means to prevent adult diabetes, cardiovascular and cerebrovascular diseases, and other chronic diseases [37]. Our study showed that it is very common for school-age children to have poor eating habits, especially being picky eaters (71.6%), having late-night snacks (55.1%), and eating meals at irregular times (53.8%). Skipping breakfast, choosing soft drinks to quench their thirst and picky eating were significantly correlated with the presence of overweight/obesity in children. Other studies have also showed similar results. Research by Abdullah Nurul-Fadhilah [38] et al. indicated that children and adolescents who did not eat breakfast regularly were more likely to develop abdominal and systemic obesity. A cohort study has shown that skipping breakfast can lead to elevated blood markers of insulin and low-density lipoprotein cholesterol in the body, adversely affecting individuals’ body weight and long-term metabolism [39]. Therefore, nutritional knowledge regarding the importance of breakfast should be spread so that parents and children realize the negative effects of skipping breakfast, thereby reducing the occurrence of related health problems.

The consumption of soft drinks continues to increase worldwide [40,41]. Studies have shown that the percentage of children and adolescents who consume soft drinks in China increased from 73.58% to 90.49% from 2004–2011, and the average weekly consumption increased to 1.5 L [42]. In Mexico, the energy intake of soft drinks increased significantly from 1999 to 2012, contributing to 9.8% of the total daily energy intake by 2012 [43]. An excess consumption of soft drinks can induce obesity, dental caries, and shortened sleep periods, which can lead to severe damage to the body’s organs and increase the risk of diabetes, hypertension, and all-cause mortality [44,45,46]. A meta-analysis suggested that for every 12 ounces of soft drinks children consumed every day during a year, their BMI increased by 0.06 unit [47]. The European Academy of Pediatrics and the Child Obesity Group strongly proposed limiting the intake of soft drinks by children and adolescents and promoting the consumption of drinking water and other unsweetened drinks [48]. Eating while watching television decreases an individual’s overall dietary quality. Children tend to eat unconsciously when watching television; thus, they consume soft drinks and high-fat and high-energy food more frequently than they do fruits and vegetables [49,50]. Moreover, physical exercise can be replaced by sitting and watching television, which reduces an individual’s level of physical activity and energy consumption and leads to weight gain. Moreover, another finding in our study was that the percentage of children who were picky eaters was smaller than that of children who were not picky eaters among the overweight/obese children. Those with picky eating habits selected foods with relatively less nutrition and energy, and their BMI and body fat percentage were lower [51]. Nevertheless, other studies have reported the opposite results. Finistrella [52] found that overweight and obese children were significantly more selective about their food than were normal-weight children. Long-term picky eating (an inadequate intake of protein, dietary fiber, fruits, and vegetables) and a smaller intake of iron and zinc can lead to inadequate nutrition and energy needs, which can have negative effects on individuals’ health (e.g., growth disorders and nutritional deficiencies) [53]. Therefore, long-term longitudinal cohort studies and randomized controlled trials are needed to explore the relationship between picky eating and children’s weight status and growth.

Moreover, in this study, we also investigated snack consumption in school-age children. We found that the most commonly eaten snacks were high-energy “restricted snacks” such as: biscuits/cakes (48.4%), candy/chocolate (40.1%), spicy snacks (21.4%), soda/drinks (20.6%), and puffed food (20.0%). These results are consistent with the results of Ma Guansheng et al. [54] and Liu Jing et al. [55]. The percentage of girls consuming candy/chocolate and biscuits/cakes was slightly higher than that of boys. However, boys were more likely to choose puffed food, soda/drinks, or spicy snacks. This preference could be due to the difference in eating habits between genders. Girls prefer sweets, while boys prefer soft drinks and meat products. Our study also indicated that eating puffed food was a risk factor for overweight/obesity. “Snacks” are considered a major cause of overweight and obesity [56]. Teenagers who were overweight and obese ate more snacks a day and consumed higher levels of added sugar, saturated fat, and sodium from snacks [57]. It is very common for children and adolescents in Europe and the United States to eat snacks [56]. In the United States, since 1977–1978, the percentage of children who eat snacks during a day increased from 74% to 98% by 2003–2006 [58]. Kerr MA [59] et al. analyzed the types, frequencies, and portions of snacks consumed by adolescents in Britain, in 1997, and Northern Ireland, in 2005, and found that the amount and frequency increased significantly, with soft drinks remaining the most popular. In Europe, the children’s favorite snacks were potato chips, crispy snacks, candy or chocolate, yogurt, and soft drinks [60]. According to a survey, in 1998 and 2008, the percentage of snacks consumed by children aged 8–14 years in cities in China was consistently higher than 98%, with an obviously decreased intake of vegetables and fruits [61]. Therefore, school-age children should eat moderate amounts of different kinds of snacks to meet their dietary needs for physical growth and development. It is recommended that children choose milk, fruits, vegetables, nuts, etc. instead of high-fat and high-energy “restricted snacks”, such as candy and fried food. (“Restricted snacks: Snacks consumption guide for Chinese children and adolescents” regarded high-sugar, high-salt, and high-fat snacks as snacks whose amount of consumption should be restricted. It is recommended that these snacks are not eaten more than once a week.).

Considering that poor eating habits have a direct impact on the weight status and growth of school-age children, we suggest that the following measures are taken. First, health education should be increased, and public health awareness should be enhanced due to the potential negative consequences of obesity in school-age children and caregivers. Targeted interventions should be designed based on the particularities of school-age children’s living and learning environments. Second, children’s diets should be managed and controlled. For example, caregivers should set an example for their children and encourage them to develop good eating habits and lifestyles at home. Schools should strictly limit the amounts of snacks students can purchase and should not make unhealthy snacks available around the school. Finally, for the state or government, the food production enterprises should be supervised, and policies and regulations should be formulated to restrict unhealthy food marketing.

## 5. Strengths and Limitations

This study contributes to understand the association between eating habits and weight status in Chinese primary school-age children and helps to determining potential risk factors. We used a large sample size and assessed eating habits questionnaires which was easy to administer.

Nevertheless, there are also some limitations in this study. First, this study was a cross-sectional study that did not confirm the causal inferences. Second, because the “Classification Criteria for Body Mass Index Value of Overweight and Obesity Screening for School-age Children and Adolescents in China” did not define thinness, we used the WHO’s “Growth Reference Z Score Standard for School-age Children and Adolescents” to define thinness, which could have led to some errors regarding China’s standards. Last, we only preliminarily discussed eating habits and snack consumption. It is essential to further analyze the intake and sources of total energy and free sugar in children’s diets in the future to provide more evidence important for the development of dietary interventions that prevent obesity in children in China.

## 6. Conclusions

In this study, the prevalence of overweight and obesity in Chinese school-age children is high, and they exhibited poor eating habits. Children who skipping breakfast, being picky eaters and eating puffed food were at increased risk of being overweight and obesity. Those who were boys, only children, and living in urban area had a higher percentage of overweight and obesity. Our findings indicated that it is necessary to develop nutritional strategies to prevent childhood obesity. Longitudinal or case-control studies are necessary to further analyze the relationship between eating habits, physical activity, and weight status in Chinese children.

## Figures and Tables

**Figure 1 ijerph-17-03571-f001:**
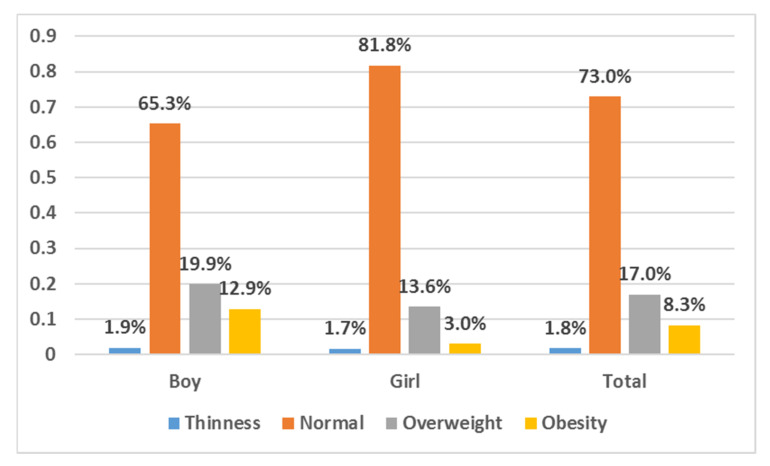
Body mass index (BMI) categories by stratified by sex (Chi-square test, *p* < 0.001).

**Figure 2 ijerph-17-03571-f002:**
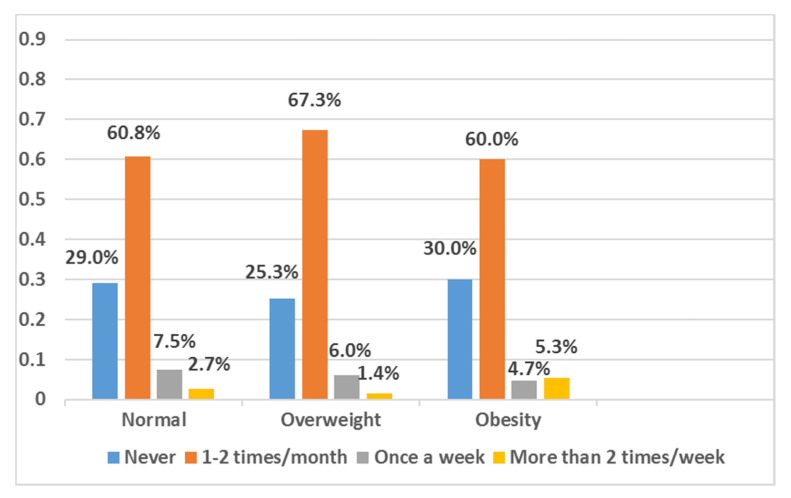
Frequency of eating fast food (KFC, McDonalds, etc.) among school-age children by BMI categories.

**Figure 3 ijerph-17-03571-f003:**
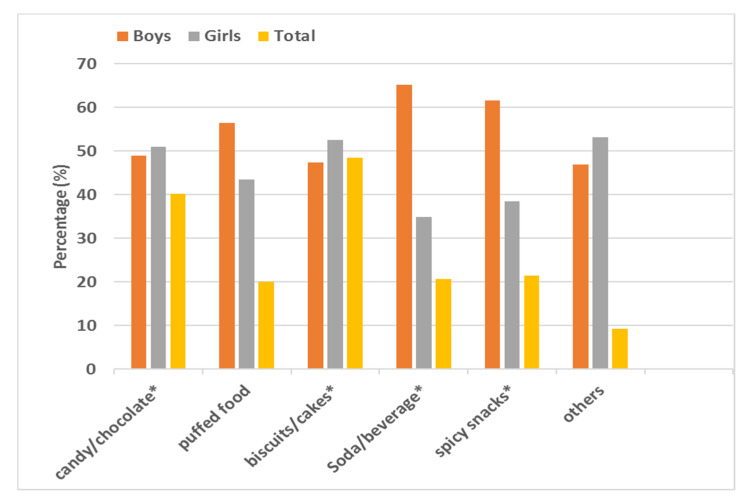
Types and percentages of snacks consumed more than three times a week by school-age children (Chi-square test, * *p* < 0.01).

**Table 1 ijerph-17-03571-t001:** Sociodemographic information of male school-age children by weight status.

	N	Normal	Overweight/Obesity	*p*-Value
		n, %	n, %	
All	1174	767 (65.3)	385 (32.8)	
Age (y)				0.043
7~9	262	185 (70.6)	73 (27.9)	
10~12	902	574 (63.6)	311(34.5)	
13~15	10	8 (80.8)	4 (10.0)	
Only child ^#^				0.011
Yes	708	442 (62.4)	253 (35.7)	
No	404	287 (71.0)	109 (27.0)	
Single-parent family ^#^				0.177
Yes	113	70 (61.9)	43 (38.1)	
No	1004	656 (65.3)	327 (32.6)	
Living area				<0.001
Urban	749	451 (60.2)	292 (39.0)	
Town	425	316 (74.4)	93 (21.9)	
Attended boarding school ^#^				<0.001
Yes	84	64 (76.2)	14 (16.7)	
No	1029	658 (63.9)	355 (34.5)	
Snack cost (RMB) ^#^				0.155
<5	713	480 (67.3)	221 (31.0)	
≥5	409	252 (61.6)	149 (36.4)	
Physical activity duration per day (hour) ^#^				0.010
≥1	624	384 (61.5)	228 (36.5)	
<1	522	365 (69.9)	147 (28.2)	

^#^ Missing data not included.

**Table 2 ijerph-17-03571-t002:** Sociodemographic information of female school-age children by weight status.

	N	Normal	Overweight/Obesity	*p*-Value
		n, %	n, %	
All	1011	827 (81.8)	167 (16.5)	
Age (y)				0.102
7~9	270	224 (83.0)	43 (15.9)	
10~12	732	598 (81.7)	121 (16.5)	
13~15	9	5 (55.6)	3 (33.3)	
Only child ^#^				0.001
Yes	495	390 (78.8)	100 (20.2)	
No	486	414 (85.2)	60 (12.3)	
Single-parent family ^#^				0.094
Yes	75	55 (73.3)	19 (25.3)	
No	913	755 (82.7)	143 (15.7)	
Living area				0.001
Urban	669	532 (79.5)	129 (19.3)	
Town	342	295 (86.3)	38 (11.1)	
Attended boarding school ^#^				0.988
Yes	49	40 (81.6)	8 (16.3)	
No	916	748 (81.7)	152 (16.6)	
Snack cost (RMB) ^#^				0.140
<5	678	573 (70.0)	105 (15.5)	
≥5	300	243 (81.0)	55 (18.3)	
Physical activity duration per day (hour) ^#^				0.521
≥1	457	371 (81.2)	76 (16.6)	
<1	533	442 (82.9)	84 (15.8)	

^#^ Missing data not included.

**Table 3 ijerph-17-03571-t003:** Differences in eating habits among the children with different weight statuses *.

	Normal	Overweight/Obesity	Total	*p*-Value
	(n = 1633)	(n = 552)	(n = 2185)	
Skip breakfast ^#^				0.032
Every day	10 (55.6)	8 (44.4)	18 (0.8)	
1–2 times/week	88 (72.7)	31 (25.6)	121 (5.6)	
3–4 times/week	78 (62.4)	43 (34.4)	125 (5.8)	
5–6 times/week	126 (67.4)	56 (29.9)	187 (8.6)	
Never	1285 (74.6)	409 (23.8)	1722 (79.2)	
Eat out ^#^				0.129
≥2 times/week	490 (74.9)	149 (22.8)	654 (30.5)	
Once a week or less	1074 (72.2)	391 (26.3)	1488 (69.5)	
What to choose to quench their thirst ^#^				0.039
Carbonated drinks (coke, sprite, etc.)	95 (69.3)	37 (27.0)	137 (6.4)	
Mineral water	164 (64.3)	86 (33.7)	255 (12.0)	
Fruit juice	108 (74.0)	35 (24.0)	146 (6.8)	
Tea drinks	89 (76.1)	26 (22.2)	117 (5.5)	
Plain boiled water	1105 (74.8)	348 (23.6)	1477 (69.3)	
Eat while watching television ^#^				0.112
Often	154 (67.0)	74 (32.2)	230 (10.7)	
Occasionally	958 (73.7)	316 (24.3)	1300 (60.3)	
Never	460 (73.4)	156 (24.9)	627 (29.1)	
Picky eaters ^#^				<0.001
Never	391 (65.5)	198 (33.2)	597 (28.4)	
Occasionally	1053 (76.1)	307 (22.2)	1384 (65.7)	
Often	98 (79.0)	22 (17.7)	124 (5.9)	
Eating speed ^#^				0.141
≤10 min	284 (70.8)	113 (28.2)	401 (18.7)	
>10 min	1286 (73.5)	428 (24.5)	1749 (81.3)	
Frequency of eating fast food (KFC, McDonalds, etc.) ^#^				0.583
Never	441 (74.6)	140 (23.7)	591 (28.5)	
1–2 times/month	923 (71.9)	339 (26.4)	1283 (61.9)	
Once a week	114 (79.2)	29 (20.1)	144 (6.9)	
More than 2 times/week	41 (73.2)	14 (25.0)	56 (2.7)	
Have late-night snacks ^#^				0.685
Never	678 (71.1)	257 (27.0)	953 (44.9)	
Occasionally	829 (74.1)	270 (24.1)	1119 (52.7)	
Often	38 (73.1)	13 (25.0)	52 (2.4)	
Eating meals at irregular times ^#^				0.782
Yes	849 (73.1)	289 (24.9)	1161 (53.8)	
No	729 (73.0)	254 (25.4)	999 (46.3)	

* Thinness and very thinness were not included. ^#^ All categories do not have the same sample sizes due to missing data.

**Table 4 ijerph-17-03571-t004:** Logistic regression analysis of the factors of overweight/obesity among school-age children.

Variables	Reference Group	Adjusted OR (95% CI)	*p*-Value
Boys	Girls	2.913 (2.239~3.790)	0.000
Urban	Town	1.958 (1.465~2.618)	0.000
Only child	No	1.370 (1.055~1.781)	0.018
Snack cost < 5 RMB/day	≥5 RMB	0.714 (0.548~0.932)	0.013
Skip breakfast	No	1.507 (1.116~2.035)	0.007
Picky eaters	No	0.559 (0.427~0.733)	0.000
Eat puffed food ≥ 3 times/week	No	1.571 (1.170~2.110)	0.003

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
