# Peer review of "Eating Habits and Their Association with Weight Status in Chinese School-Age Children: A Cross-Sectional Study"

_ijerph, 2020, doi:10.3390/ijerph17103571_

Round 1
Reviewer 1 Report
I was honored to review the manuscript entitled "Eating habits and their association with weight status in Chinese school-age children: A cross-sectional study" submitted to International Journal of Environmental Research and Public Health.
.
Taking into account the multiple studies ongoing in this field this type of study is needed. I have only few small remarks that authors should adress properly.
I recommend to accept the manuscript after minor revision.
There are only some points to correct:
- please provide clearly the aim of this study
- please provide the list of abbreviations.
- please provide the number of Ethical Approval
- In my opinion discussion is too short, so I suggest that Introduction and Discussion section needs improvement- please cite doi: 10.3390/jcm9020469. ; 10.1097/MD.0000000000014909. ; 10.20452/pamw.4426.
- In discussion please provide “study strong points” and “study limitation” section.
I recommend to accept the manuscript after minor revision.
Author Response
请参见附件。

Reviewer 2 Report
1.-What was the reason for taking a sample of 2224 students?
2.-In the introduction, it is recommended to add more references of the different topics to be addressed, such as studies currently carried out, types of food that are consumed mostly in the studied population, impact on health of obesity and how this affects the development of other chronic diseases.
3.-Figure 1 is not clear, improve image quality.
4.- The presentation of results and discussion can be put together to follow the sequence of the study, it is also recommended to add figures or tables so that the data and information analyzed can be better explained visually.
5.- With the data obtained, the conclusion must be posed in a better way for a better understanding of what you want to express.
6.- The results of Table 1 can be better presented by dividing them into boys and girls, a table for each one. And it is also recommended to put the percentage of people who are overweight or obese compared to those who have a normal weight to see the relationship there is.
7.- In Table 2, the data that most attracts attention is the Frequency of eating fast food (KFC, McDonalds, etc.). of this they could make a graph to highlight it.
8.-The subject is very interesting, however the presentation of results and how the information obtained is being explained is poor.
Reviewer 3 Report
This cross-sectional study conducted in a Chinese province aimed to determine eating habits and physical activity of 2185 school age children and their association with BMI.
Several drawbacks limit the interest of the study.
Major comments
- This study does not provide contemporary data, since it was conducted 8 years ago
- As cross-sectional study, it only provide associations and do not add relevant data to predictive data from well-designed longitudinal studies addressing the same subject
- The Introduction is to extensive, deviating from the focus when addressing worldwide prevalence (US, Crete) of overweight and obesity in children and adolescents, instead of focusing on data from China.
- Most of the questionnaires was designed to obtain a dichotomous response (yes/no), instead of more informative quantitative data, such as the time spent eating while watching TV, and the frequency of skipping breakfast, choosing soft drinks, picky eating, late-night snacks, and eating meals at irregular times.
- Accuracy of height measurements is of utmost importance since any error in measurement is amplified when its value is squared for BMI calculation. It should be specified if any measure of inter-observer variation has been obtained. If not, this is a weakness and should be acknowledged as a limitation.
- The Conclusions should only include results obtained from the logistic regression analysis.
Minor comments
- The objective (lines 62-63) should be better stated, including the record of physical activity and its association with BMI.
- In Discussion the p-values should not be included; instead, it is understandable if stating that associations were significant
- Many statements need support bay a reference, such as those included in lines 55-56, 173-174, 175-176, 187-191.
- Line 272: in fact, a cross-sectional study is able to determine a relationship (association), but not a cause-effect relationship
Round 2
Reviewer 3 Report
The authors made a great effort to revise the manuscript, which is improved. Some unmodifiable weaknesses (eg, not contemporary cross-sectional study) remain. Nevertheless, factors associated to childhood obesity identified in the great sample size analyzed may contribute to develop dietary preventive strategies for Chinese children.